# Polycation-Intercalated MXene Membrane with Enhanced Permselective and Anti-Microbial Properties

**DOI:** 10.3390/nano13212885

**Published:** 2023-10-31

**Authors:** Jie Yang, Shilin Zhu, Hongli Zhang

**Affiliations:** 1School of Materials Science and Engineering, Xi’an Polytechnic University, Xi’an 710048, China; 2School of Materials Science and Chemical Engineering, Xi’an Technological University, Xi’an 710021, China

**Keywords:** Ti_3_C_2_T_x_, MXene membrane, nanofiltration, two-dimensional membrane, anti-biofouling, PDDA

## Abstract

Two-dimensional (2D) nanomaterial-based membranes feature attractive properties for molecular separation and transport, which exhibit huge potential in various chemical processes. However, the low permeability and bio-fouling of the MXene membrane in water treatment become huge obstacles to its practical application. Herein, a highly permselective and anti-bacterial 2D nanofiltration membrane is fabricated by intercalating a polycation of polydiallyldimethylammonium chloride (PDDA) into the Ti_3_C_2_T_x_ MXene laminar architecture through a facile and patternable electrostatic assembly strategy. As a result, the as-fabricated Ti_3_C_2_T_x_/PDDA composite membrane exhibits higher water permeance up to 73.4 L m^−2^ h^−1^ with a rejection above 94.6% for MgCl_2_. The resultant membrane simultaneously possesses good resistance to swelling and long-term stability in water environments, even after 8 h. Additionally, the Ti_3_C_2_T_x_/PDDA membrane also demonstrates a high flux recovery ratio of nearly 96.1% to bovine serum albumin proteins after being cleaned. More importantly, the current membrane shows excellent anti-adhesive and anti-microbial activity against Gram-negative *Escherichia coli* (*E. coli*) and Gram-positive *Staphylococcus aureus* (*S. aureus*), with inhibition rates of 90% and 95% against *E. coli* and *S. aureus*, respectively. This holds great potential for the application of the polyelectrolyte-intercalated MXene membrane in serving as a promising platform to separate molecules and/or ions in an aquatic environment.

## 1. Introduction

Accompanied by overpopulation and accelerated urbanization, serious environmental pollution and the extreme shortage of water resources have posed a great threat to human survival and development. To mitigate the environmental risk and provide more clean water for the public, several strategies have been developed that aim to treat wastewater and reuse it. Membrane technology is regarded as an efficient approach due to the robust performances of membranes in reclamation, water treatment, desalination, and chemical processing [1]. As one of the membrane separators, a nanofiltration membrane can reject solutes that are nominally 1 nm in dimension with a molecular weight cut-off ranging from 200 to 1000 Da. In addition, a promising usage value is exhibited owing to the merits of high fluxes, a low operating pressure, a high amount of rejection to divalent ions, a low retention of monovalent ions, and low operation costs [2,3]. 

In recent years, two-dimensional (2D) materials have gradually become an ideal choice for the fabrication of nanofiltration membranes on account of their chemical properties and unique morphology [4]. Compared with traditional polymer membrane materials, the unique mono-atomic thickness of 2D nanomaterials enables the formation of an ultrathin membrane-separating layer, which minimizes transport resistance and maximizes permeation flux, thus offering ultimate separation capabilities. Over recent years, various two-dimensional nanomaterials, such as GO [5], MXene [6,7], and MoS_2_ [8], have been selected as superb building blocks for constructing nanofiltration membranes. Among them, MXene is widely researched because of its outstanding physical and chemical properties, such as better rigidity, abundant surface groups, and natural hydrophilicity. Additionally, MXenes can also be easily prepared in a high yield using wet chemical methods from inexpensive precursors and are quickly processed via multiple methods (including filtration, spray coating, printing, etc.) to form various functional devices. The MXenes could also be modulated by other functional species and materials that enable the utilization of MXenes as absorbed, separative, and photocatalytic membranes. Based on the above physiochemical characteristics, the resultant MXene nanofiltration membrane can be widely used in waste water decontamination, such as for the effective removal of ions, heavy metals and dye molecules [9,10]. However, water molecules can be spontaneously absorbed into the interlayer of Ti_3_C_2_T_x_ nanosheets, resulting in swelling and poor stability [11,12,13]. At the same time, Ti_3_C_2_T_x_ nanosheets tend to form dense structures through face-to-face stacking and aggregation in thermodynamics. Non-selective defects caused by the inherent repulsion force between negatively charged Ti_3_C_2_T_x_ nanosheets limit the exposed surface area and water penetration [14,15].

To solve the above problems, many strategies have been proposed, such as nanoparticle/ion embedding, self-crosslinking, and covalent cross-linking [16,17]. For instance, Ding et al. employed Fe(OH)_3_ colloid nanoparticles for intercalation to expand the interlayer channel size [18]. The positively charged Fe(OH)_3_ colloid intercalated into negatively charged MXene nanosheets via the electrostatic force and formed an extended nanochannel, which can increase the volume of the interlayer cavity and boost the membrane permeation flux. Although the insertion of nanoparticles into a Ti_3_C_2_T_x_ interlayer can effectively prevent the stacking and aggregation behavior of the MXene membrane, they may also be separated from the Ti_3_C_2_T_x_ membrane in the operation process and further aggravate environmental pollution, resulting in the problem of them being difficult to recovery [19]. The importing of crosslinkers and polymer molecules can fix the layer spacing and mitigate the swelling issue of Ti_3_C_2_T_x_ membranes; however, the selectivity of small-sized hydrated molecules and/or ions can also be sacrificed to some extent [20]. Meanwhile, the crosslinking usually accompanies a relatively complex operation process [16]. Therefore, it is a pressing requirement for the fabrication of a novel MXene nanofiltration membrane with long-term stability and superior separation performance.

Membrane fouling, especially biofouling, seriously limits the overall performance of nanofiltration membranes, including their rejection, flux, and durability [21]. The main reason for biofouling is that the accumulation and adhesion of bacteria on the membrane surface lead to the growth of microbial cells, which can proliferate and gradually form a biological layer containing microbial cells and extracellular polymers [22]. The biological layer ultimately destroys the membrane structure and greatly reduces the selectivity, thus increasing operation costs and shortening the service life of the membrane. Therefore, engineering a bacterial-resistant MXene nanofiltration membrane that is chemically stable and maintains a high water permeability is urgently desired for the water purification and desalination industries.

It has been believed that adjusting the surface charge of membrane materials is a feasible method to regulate adhesion resistance and prevent microbial contamination [23]. An electrostatic attraction exists between membrane surfaces, and the negatively charged surface of bacteria can disrupt or impose a charge imbalance, which would impel the breakdown of cell membranes, resulting in a leakage of the cellular content, and eventually leading to the death of microbes [19,24]. In this work, we propose to simultaneously enhance the separation performance and anti-biofouling properties of a nanofiltration membrane by importing a polycationic electrolyte of PDDA in the MXene separation layer. On the one hand, the positively charged PDDA molecular can serve as pillars to regulate the water transport channel, and further as crosslinking agent through the electrostatic interaction to inhibit the swelling phenomenon (As shown in Figure 1). Also, PDDA molecular can further increases the hydrophilicity of the MXene membrane, which is expected to accelerate the selective permeation of water molecular and anti-adhesive properties [20]. Moreover, the polycation of PDDA can disrupt bacterial cytomembrane kill bacteria, and endow the MXene/PDDA composite membranes with antibacterial ability. As a result, the prepared Ti_3_C_2_T_x_/PDDA composite membrane delivers a high pure water flux (73.4 L m^−2^ h^−1^) and a rejection above 94.6% for MgCl_2_. Additionally, the anti-pollution performance of Ti_3_C_2_T_x_/PDDA membrane to bovine serum albumin protein is investigated, and a high flux recovery ratio of nearly 96.1% is achieved after being cleaned. The inhibition rate against Gram-negative *Escherichia coli* (*E. coli*) and Gram-positive *Staphylococcus aureus* (*S. aureus*) reaches up to 90% and 95%, respectively. This work provides an innovative conception and theoretical reference for the designing of a 2D nanofiltration membrane, which can hopefully be used in wastewater treatment.

## 2. Experimental Section

### 2.1. Materials

MAX (Ti_3_AlC_2_) was brought from 11 Technology Company (Jilin, China). Polyethersulfone (PES) base membrane (with a pore size of 0.2 µm and a diameter of 50 mm) was provided by Jinteng Experiment Equipment Co., Ltd. (Tianjin, China). Hydrochloric acid (HCl), lithium fluoride (LiF, 99%), bovine serum albumin (BSA), poly(diallyldimethylammonium chloride) (PDDA), penicillin G potassium salt (PG, 98%), tetracycline (TET, CP), and erythromycin (EM, 98%) were purchased from Aladdin Biochemical Technology Company (Aladdin, Shanghai, China). Sodium chloride (NaCl), magnesium chloride (MgCl_2_), sodium sulfate (Na_2_SO_4_), and magnesium sulfate (MgSO_4_), were obtained from the Damao Chemical Reagent Factory (Damao, Tianjin, China). All of the above regents were directly used without further purification. *E. coli* and *S. aureus* strains were purchased from Beijing Biotechnology Company (Beijing, China). Deionized (DI) water was used in all experiments.

### 2.2. Preparation of Ti_3_C_2_T_x_-Polyeletrolytes Membrane 

As schematically depicted in Figure 1, the Ti_3_C_2_T_x_/PDDA polyelectrolyte composite membrane was fabricated through a simple vacuum filtration strategy. The few-layered Ti_3_C_2_T_x_ MXene was prepared by adapting a typical selective etching and ultrasonic exfoliation of the Ti_3_AlC_2_ MAX phase by referencing previous work [11]. To form a Ti_3_C_2_T_x_-polyeletrolytes membrane, 1.0 mg mL^−1^ Ti_3_C_2_T_x_ suspension was mixed with different mass rations of PDDA solution and ultrasonic treated for 30 min for homogenization. Then, the obtained Ti_3_C_2_T_x_/PDDA mixture solution was vacuum-filtered on a commercial PES-based membrane. After drying in air for 1 h, the Ti_3_C_2_T_x_/PDDA membrane was obtained. The Ti_3_C_2_T_x_/polyeletrolyte membrane prepared with different PDDA contents were respectively denoted as MP1, MP2, MP3 and MP4. For comparison, the pure Ti_3_C_2_T_x_ MXene membrane was also prepared by using the same method.

### 2.3. Membrane Characterization 

The crystal phase was examined by X-ray diffraction (XRD D2 Advance, AXS, Karlsruhe, Germany) with Cu Kα radiation. The chemical composition and state are analyzed by using X-ray photoelectron spectroscopy (Thermo Kalpha, Thermo ESCALAB 250XI, Thermo Fisher Scientific Inc., Waltham, MA, USA) and Fourier transform infrared (FTIR IS50, Thermo Fisher Scientific Inc., Waltham, MA, USA) spectroscopy. The morphologies and microstructures were investigated by scanning electron microscopy (ZEISS Sigma 300, Tokyo, Japan) and transmission electron microscopy (JEM2010, JEOL, Ltd., Tokyo, Japan). The Atomic Force Microscope (AFM, Dimension ICON, Bruker, Germany ) was used to study the surface roughness and membrane phase diagram. The water contact angle was measured by employing video-based contact angle instrument (OCA15EC, Dataphysics, Stuttgart, Germany) and zeta potential of the membrane was analyzed with a Zetasizer (Malvern Nano ZS90, Malvern, UK). 

### 2.4. Measurements of Separation Performance and Anti-Biofouling Activity

The separation performance was evaluated by using a cross-flow filtration experimental device. The membrane samples with an effective area of 8.04 cm^2^ were firstly preloaded with pure water at 0.2 MPa until a steady flux state was reached, then the water flux was measured. All performance tests were conducted at an operation pressure of 0.6 MPa and a temperature of 25 °C. To evaluate the antifouling ability, BSA (1.0 g L^−1^) and MgCl_2_ (2000 ppm) mixture aqueous solution was chosen to simulate the contamination conditions. Firstly, the stable water permeability (*J*_w1_) of membranes were measured by using MgCl_2_ solution at 0.6 MPa for 1 h. Then, the permeability (*J*_p_) was measured by filtration using BSA aqueous at 0.6 MPa for 8 h. Finally, the membranes was washed with deionized water for 0.5 h and the stable water permeability (*J*_w2_) was measured again with MgCl_2_ solution at 0.6 MPa for 5 h. The permeation flux (*J*) and rejection (*R*) were calculated using Equations (1) and (2):(1)$$J=\frac{V}{A\times t}$$
(2)$$R=\left(1-\frac{Cp}{Cf}\right)\times 100\%$$
where *V* is the volume of permeate pure water (L), *A* is the effective area of the membrane (m^2^), *t* is the permeation time (h), *C*_p_ is the permeate concentration and *C*_f_ shows the feed concentration. The concentration of inorganic salt was determined by measuring the ionic conductivity with a conductivity meter, and the concentration of antibiotic solution was measured by liquid chromatography (HPLC, Thermo U3000, Thermo Fisher, Waltham, MA, USA). Antifouling values of flux recovery ratio (*FRR*) and total flux decline ratio (*DR*) were calculated through Equations (3) and (4).
(3)$$FRR\%=\frac{J_{W2}}{J_{p}}\times 100\%$$
(4)$$DR\%=\frac{J_{W1}-J_{P}}{J_{W1}}\times 100\%$$

### 2.5. Anti-Bacterial Activity Test

For inhibition rate test, the strains of Gram-negative bacteria *Escherichia coli* (*E. coli*) and Gram-positive bacteria *Staphylococcus aureus* (*S. aureus*) were, respectively, picked by inoculation ring, and then placed in 250 mL liquid medium and cultured in a constant temperature chamber at 37 °C for 24 h to obtain the first generation of bacteria liquid. Then, 1.0 mL of first generation of bacteria liquid was added to 100 mL liquid medium and cultured for 24 h at 37 °C, and the second generation of bacteria liquid was acquired. To test anti-bacterial properties, the membrane samples were fully immersed in 10 mL of second-generation bacteria suspension (blank control group was added as comparison) and cultured at 37 °C for 24 h. The OD values were measured by UV–Vis spectrophotometer, and the antibacterial rate (*k*%) was calculated by using Equation (5), where OD_C_ is the absorbance of control group, and OD_S_ represents the absorbance of experimental group.
(5)$$k\%=\left(1-\frac{{OD}_{S}}{{OD}_{C}}\right)\times 100$$

Disc inhibition zone assay. To further explore the antibacterial activities of membrane samples, a disc inhibition zone test was performed to determine the inhibition zone [25,26]. *E. coli* and *S. aureus* were respectively picked by inoculation ring and placed in 250 mL liquid medium and cultured in a constant temperature chamber at 37 °C for 24 h to obtain the first-generation bacteria liquid. Then, 1.0 mL of first generation of bacteria liquid was added to 100 mL liquid medium and cultured for 24 h at 37 °C, the second-generation bacteria liquid was acquired. A sterile cotton swab was then dipped into *E. coli* and *S. aureus* suspension and spread on the agar plates. The membranes were cut into discs with a diameter of 1.0 cm, which were placed on the surface of the agar plates already inoculated with bacteria. After stationary incubation at 37 °C for 24 h, the inhibition zone for each sample was determined.

For the SEM images, the membrane samples exposed to bacteria were washed with phosphate-buffer saline and DI water to remove non-adherent bacteria, and then successively washed with different percentages of laboratory-grade ethanol to enable clear imaging of the healthy viable bacterial cells and damaged bacterial cells on the tested membranes. Scanning electron microscopy (ZEISS Sigma 300, Tokyo, Japan) was used for capturing images of the live/dead bacterial cells on the membranes.

## 3. Results and Discussion

### 3.1. Materials and Membrane Characterization

The Ti_3_C_2_T_x_ nanosheets were obtained by employing selective etching (in the HCl-LiF system) and exfoliation processes of the Ti_3_AlC_2_ precursor. Appendix A exhibits the XRD patterns of the few-layered Ti_3_C_2_T_x_ MXene and Ti_3_AlC_2_. It can be seen two characteristic peaks of Ti_3_AlC_2_ located at 38.5° and 9.8° completely disappear and a significant diffraction peak of 6.5° emerges in the few-layered Ti_3_C_2_T_x_ MXene. This result is consistent with previous work and suggests the Al layer is completely removed from the MAX precursor, and the interlayer space of Ti_3_C_2_T_x_ MXene has been expanded [27,28]. A significant Tyndall scattering effect can be observed while a beam of laser goes through the Ti_3_C_2_T_x_ MXene suspension (Appendix A), indicating excellent water dispersity and stability of the MXene colloid suspension. In addition, the morphology of few-layered Ti_3_C_2_T_x_ nanosheets was investigated by using the TEM technique. Several near-transparent sheets with a lateral dimension ranging from hundreds of nanometers to several micrometers can be seen (Appendix A), demonstrating the few-layered Ti_3_C_2_T_x_ MXene nanosheets exhibit a relatively thin thickness. 

The fabrication procedure of the MXene/PDDA composite membrane is depicted in Figure 1. The few-layered Ti_3_C_2_T_x_ MXene is mixed with PDDA polyelectrolyte solution and then vacuum-filtered on the PES base membrane, and the MXene/PDDA composite membrane is acquired. In the experimental process, it is found that floccules appear while dropping the PDDA solution into MXene dispersion because of the electrostatic interaction (Appendix A), which is beneficial to form the PDDA intercalation into MXene layer space. The chemical composite of MXene/PDDA membrane was characterized by FTIR. As shown in Appendix A, the individual MXene membrane features two characteristic peaks of 3430 cm^−1^ and 1654 cm^−1^, which respectively, correspond to the stretching vibration of -OH and C-O groups of MXene. After functionalization with PDDA polyelectrolyte, several new peaks located at about 2089 cm^−1^, 1078 cm^−1^ and 1396 cm^−1^ emerge in FTIR spectra of the MXene/PDDA membrane, which were attributed to the C-N bond and -CH_3_ of the methylammonium groups, respectively [29,30]. Additionally, the redshift of the -OH group from the 3430 cm^−1^ to 3256 cm^−1^ peak reveals the formation of hydrogen bonds between the -OH on MXene and N atoms of PDDA polymer chains [31]. XPS characterization was applied to further verify the chemical state of the MXene/PDDA membrane. According to the XPS survey spectra (Figure 2a), it is obvious that a weak enhancement of intensity for the N peak can be observed in the MXene/PDDA membrane, which may be caused by the importing of PDDA molecular, indicating the successful combination of MXene and PDDA Figure 2b shows the C1s spectra of the MXene membrane belonging to the C-Ti, C-C, C-O and C-Ti=O bands, respectively. However, a new peak belonging to C-N at ~285.5 eV was detected from the MXene/PDDA membrane (Figure 2c) [32]. It probably is generated from the chemical bonds between the amines of PDDA and the oxygen-containing groups of MXene. We further analyze the high-resolution O1s XPS spectrum of the MXene/PDDA membrane and compare it with that of the MXene membrane (Appendix A). The binding energy of Ti-O and C-Ti-T_x_ (T is O and OH) slightly decreased because of the interface electron transfer caused by the hydrogen bonding between N atom of PDDA chains and the hydroxyl groups of the MXene surface. These results are in good agreement with the FTIR analysis. The N1s spectrum of MXene/PDDA membrane shows two peaks at 399.3 eV and 402.1 eV (Figure 2d), respectively, attributing to the charged and uncharged quaternary ammonium fractions, further revealing that PDDA molecular could enter the interlamellar space of Ti_3_C_2_T_x_ MXene [33]. The interlayer space of MXene nanofiltration membrane with various PDDA content was tested with XRD and calculated through Bragg’s equation (2dsinθ = nλ); the results are shown in Appendix A. The (002) peak of MXene is left-shifted from 6.15° to 5.58°, indicating the interlayer space of adjacent Ti_3_C_2_T_x_ nanosheets increases from 14.7 Å for pure MXene membranes to 16.4 Å for MP4 membrane.

Subsequently, the surficial and cross-sectional morphology were investigated. Figure 3a,c respectively, exhibit the surface morphologies of the pure MXene membrane and MXene/PDDA membrane. Similar to some other as-reported 2D membranes, the pure MXene membrane and MXene/PDDA membrane display a typical structure of corrugation (Figure 3a). The cross-sectional SEM images of the pure Ti_3_C_2_T_x_ membrane and MXene/PDDA membrane indicate a distinct difference, as observed in Figure 3b,d. For the individual MXene membrane, the whole separating layer corresponds to several MXene nanosheets in an ordered manner, and the thickness of the MXene membrane is about 350.2 nm, which provides a clear 2D channel for the transport of water molecules [34]. After importing PDDA molecular, more folds appear on the MXene membrane surface and a slight increas emerges in the thickness of the separative layer (Figure 3d,e), suggesting that the nanosheets may stack in more disordered and add more “non-selective defects” under the electrostatic interaction, which will effectively promote the passage of target molecules. Compared to the pure MXene membrane, the MP3 membrane possesses a larger thickness (379.6 nm) and forms more additional “membrane channels” [20]. It is known that the permeation paths of the 2D membrane are determined by the adjacent nanosheets. Thus, the additional “membrane channels” contribute to the permeability of the membrane. Furthermore, AFM was used to probe the fine microstructure and roughness of the membrane surface. As shown in Figure 3c,f, the average roughness (Ra) of MXene and MP3 membranes are 59.4 nm and 67.4 nm, respectively. The MXene polyelectrolyte membrane features a larger rough surface, which suggests the polycation functionalizes the negatively charged MXene nanosheet and consequently modulates the stacking behavior through electrostatic attraction. Also, the enhanced roughness means a larger penetration area, which is expected to deliver a better water separation performance.

### 3.2. Membrane Permselectivity Properties

In order to evaluate desalination properties, cross-flow NF equipment was adopted to measure the water permeability and salt rejection of the membrane samples. Here, 2000 ppm of Na_2_SO_4_, MgSO_4_, NaCl and MgCl_2_ aqueous solution were chosen as feed solutions. Firstly, the separation performances of pure Ti_3_C_2_T_x_ MXene membranes with different Ti_3_C_2_T_x_ content toward Na_2_SO_4_ salts solution were investigated. It is clearly observed from Appendix A, that the pure Ti_3_C_2_T_x_ membrane (with a thickness of ~350.2 nm) shows a high water permeance (70.0 L m^−2^ h^−1^) and a relatively high rejection of Na_2_SO_4_ (31.6%). Subsequently, the effects of PDDA content and molecular weight on the permselectivity properties of MXene/PDDA composite membranes were explored, and the results are shown in Appendix A. In comparison, for the PDDA-intercalated Ti_3_C_2_T_x_ membrane containing similar Ti_3_C_2_T_x_ contents, the water flux for all investigated inorganic salts is improved significantly, although the rejection remains at a satisfactory level. To be specific, the water permeability of the MP3 membrane increases to 73.4 L m^−2^ h^−1^, while achieving a rejection of 94.6% for MgCl_2_. The improvement of water permeability can be ascribed to the increased surface hydrophilicity and the expanded interlayer space of Ti_3_C_2_T_x_ nanosheets. The increment of hydrophilicity was evidenced by a change of water contact angle from 54.4° to 33.9° (Figure 4a). Highly wetted surfaces would accelerate the adsorption and transport of water molecules through the membrane channel. Moreover, adjacent Ti_3_C_2_T_x_ nanosheets could be effectively separated though the intercalation of PDDA, and form a relatively wide water transport nanochannel, thus delivering higher water permeance. The improved surface hydrophilicity of the MXene/PDDA membrane would also exhibit a positive effect on the subsequent anti-biofouling capability.

Meanwhile, it is further observed in Figure 4b,c, that the salt rejection of pure Ti_3_C_2_T_x_ MXene membrane decreases in the order of R(Na_2_SO_4_) > R(MgSO_4_) > R(NaCl) > R(MgCl_2_). while for the Ti_3_C_2_T_x_/PDDA membrane, the reduction of rejection is as follows, R(MgCl_2_)> R(MgSO_4_) > R(NaCl) > R(Na_2_SO_4_). It is believed the pure Ti_3_C_2_T_x_ membrane surface terminates with -F, -OH, and -O groups, and thus, is typically more negatively charged. As explained by the Donnan exclusion theory [34,35], the negatively charged Ti_3_C_2_T_x_ MXene membrane tends to reject multivalent anions (SO_4_^2−^) with higher co-ion charge and permit multivalent cations (Mg^2+^) with higher counter-ion charge. To gain an understanding of the variation of membrane surface charge, the Zeta potential of Ti_3_C_2_T_x_ and Ti_3_C_2_T_x_/PDDA membrane in the pH range from 3.0 to 10.0 were measured. From Figure 4d, the pure Ti_3_C_2_T_x_ membrane exhibits a Zeta potential of −125 mV at a pH of 7.0. By comparison, the surface of the Ti_3_C_2_T_x_/PDDA membrane shows more positive charge over the full-ranged pH values, which is caused by the introduction of PDDA polycation. Consequently, the salt rejection rate of the membrane is determined by the synergistic effect of Donnan exclusion and pore size (steric hindrance) [35].

Additionally, the separation performance of Ti_3_C_2_T_x_/PDDA membrane for several antibiotics of penicillin (PG, 1.4 nm × 0.5 nm, M_W_ = 334.4 g mol^−1^), tetracycline (TET,1.3 nm × 0.7 nm, M_W_ = 444.4 g mol^−1^), and erythromycin (EM, 1.2 nm × 1.0 nm, M_W_ = 733.94 g mol^−1^) was assessed. 250 ppm of antibiotic aqueous solution is served as the feed solution. As observed, the Ti_3_C_2_T_x_/PDDA membrane can effectively eliminate target antibiotics molecules (Figure 4e), and a significantly higher water permeance with comparable rejection was achieved. More concretely, the water flux towards PG, TET, EM are 70.8, 78.5, and 97.2 L m^−2^h^−1^, and the corresponding rejections are 84.2%, 91.5% and 95.8%, respectively. As the molecular size of antibiotics increases, the water permeances slightly decrease, and the rejection increases. The reason may be explained by the size effect of solutes, and Ti_3_C_2_T_x_/PDDA could be regarded as a typical size-selective molecular sieving membrane. Further, as exhibited in Figure 4f, the Ti_3_C_2_T_x_/PDDA membrane shows outstanding stability concerning water permeance and antibiotic rejection during a continuous 8 h filtration experiment, which suggests the stable nanochannel structure and non-swelling ability of Ti_3_C_2_T_x_/PDDA membrane. Those results demonstrate that the MXene/PDDA composite membrane is promising to be used for many applications such as desalination, elimination of antibiotics and industrial wastewater treatment.

### 3.3. Membrane Antifouling Properties

During the operation process, the membrane surface is easy to be contaminated due to the adsorption between the membrane and some organic pollutants, which usually leads to poor permselectivity performance, high energy consumption and shortened membrane life [36,37]. Therefore, good contamination resistance is essential for the practical application of membranes. Based on this, the antifouling performance of MXene and MXene/PDDA membranes was measured by separating bovine serum albumin (BSA) from an aqueous solution. Then, the antifouling performance of membranes was evaluated by calculating the *DR* and *FRR* values. As shown in Figure 5a, the membrane flux decreases rapidly when 1.0 g L^−1^ BSA aqueous solution is used instead of pure water. This is mainly caused by concentration polarization, formation of filter layer (reversible pollution) and adsorption of fouling on the membrane surface/pore (irreversible pollution) [38]. Immediately after pollution, the membrane was washed with pure water and then the water flux of both samples would be restored. It is important to note that reversible pollutants can be eliminated by backwashing, but irreversible fouling can only be eliminated by chemical cleaning. Therefore, the latter is the main reason for shortening membrane life and increasing operating costs. According to Figure 5b, the flux decay rates (DR) of the MXene nanofiltration membrane and MP3 nanofiltration membrane were, respectively, 36.8% and 19.5%, and the *FRR* after cleaning were 85.3% and 96.1%, respectively. This result shows that antifouling performance is effectively improved while the MXene membrane functionalized by PDDA polyelectrolyte. Because of the formation of hydrated layer and strong electrostatic repulsion on the MP3 membrane surface, it is not easy to form a pollution layer for organics [39,40]. Meanwhile, compared to pure MXene membranes, the MP3 nanofiltration membranes possess a larger surface roughness, so the accumulated organic pollution layer on the surface is looser and easier to wash off and antifouling performance is increased.

### 3.4. Membrane Antibacterial Performance

Microorganisms are another important polluter of the membrane surface, which can also result in the degradation of the separation performance and severely shorten the server life of membrane materials. In view of that, two bacteria, *E. coli* and *S. aureus* have been selected as model microorganisms to estimate the anti-biological fouling properties of the MXene/PDDA membrane. Firstly, the inhibition rates of pure MXene membrane and MP3 membrane were measured and the PES base membrane was set as control. Figure 6a shows the antibacterial activity plot of different membranes while the *E. coli* and *B. subtilis* grown in different membranes-bacteria mixed culture medium. The experiment results show the inhibition rate is enhanced for both *E. coli* and *B. subtilis* while importing PDDA into the MXene separation layer. It is noted the commercial PES base membrane is not equipped with antibiosis, so a negligible inhibition rate is obtained. In particular, the MXene membrane exhibits inhibition efficiency of 20% and 48% against *E. coli* and *S. aureus.* And the MP3 membrane inhibits *E. coli* and *S. aureus* with 90% and 95% efficiency, respectively. These results demonstrate that the MP3 membrane exhibited excellent antibacterial performance against *E. coli* and *S. aureus* due to the synergistic antibacterial activity of MXene and PDDA. The monitoring of the formation of the bacteriostatic zone shows that, the width of the antibacterial band for both *E. coli* and *S. aureus* is significantly larger for MP3 membranes on the plate compared to the MXene membrane (Figure 6b,c). At the same time, accompanyied with the increment of PDDA content, the antibacterial zone gradually increases. This confirms that the presence of PDDA indeed significantly improves the anti-microbial capacity of the composite membranes.

To evaluate the interaction of the membrane with bacteria, the growth status of bacteria (*E. coli* and *S. aureus*) on the membrane surface was investigated by SEM. As displayed in Figure 6d–h, the surfaces of PES, PES-MXene and PES-MP3 membranes show different inhibitory effects on adhesion behavior of *E. coli* and *S. aureus*. The MP3 membrane presents the highest anti-adhesion behavior for bacteria cells. Only a few *S. aureus* bacteria can be seen and almost no *E. coli* cellular structures are presented on the MP3 membrane surface. The anti-adhesion inhibition ability of MXene and MP3 may be attributed to the indirect bactericidal process [41]. The antimicrobial mechanism is believed to be conducted by electrostatic attraction between negatively charged phospholipid bilayer of bacteria or organism cells membrane and the chemical species on the membrane. This interaction would impel the breakdown of cell membranes and eventually lead to the death of microbes [42]. In addition, thanks to the hydrophilicity of PDDA, the increased hydrophilicity of the MXene/PDDA composite membrane is another possible reason for the reduction of bacterial adhesion.

Furthermore, the anti-biological fouling properties of MXene/PDDA composite membrane for real wastewater were also explored. As displayed in Appendix A, we collected wastewater from a mildly polluted river, and then the antibacterial test was conducted in Petri dish by using the real river water as a bacterial strain. It is noted that the PES base membrane was also treated as the control. Appendix A show the antimicrobial result of the MXene/PDDA composite membrane for the real wastewater. As observed, for the river water without the MXene/PDDA composite membrane and the control, a large number of strains appear in culture dishes. However, for the MXene/PDDA composite membrane, no obvious strains were seen and an inhibition rate of approximately 95% can be achieved. The above results reflect the possibility of the MXene/PDDA composite membrane being applied as an effective and long-acting membrane material for the purification of real wastewater. Additionally, we compare the anti-biofouling behavior of MXene/PDDA composite membrane with that of several other recently reported membrane materials (Appendix A) [43,44,45,46,47,48,49,50,51]. Overall, although the water flux is not particularly outstanding, the antibacterial activity is comparable to others.

## 4. Conclusions

In summary, a PDDA polyelectrolyte functionalized MXene nanofiltration membrane with prominent separative performance and anti-biofouling properties was successfully developed through a simple vacuum filtration self-assembly strategy. Due to the import of PDDA polycation, the MXene membrane exhibits high removal capacity for inorganic salts and antibiotics. For the Ti_3_C_2_T_x_/PDDA membrane, a high water flux (73.4 L m^−2^ h^−1^) and a rejection up to 94.6% for MgCl_2_ can be achieved. While the Ti_3_C_2_T_x_/PDDA membrane is employed to remove antibiotics from the wastewater, a rejection of 95% for erythromycin is obtained. Furthermore, the anti-pollution performance of Ti_3_C_2_T_x_/PDDA membrane to bovine serum albumin protein is boosted and a high flux recovery ratio of nearly 96.1% is achieved after cleaning. Additionally, the PDDA endows the MXene membrane surface with a more positive charge and, thus, exhibits superior antibacterial properties. As a result, the antibacterial rate against Gram-negative *Escherichia coli* (*E. coli*) and Gram-positive *Staphylococcus aureus* (*S. aureus*) reaches up to 90% and 95%, and can inhibit the adhesion and growth of bacteria on the membrane surface. To sum up, this work shares an effective strategy to engineer 2D membranes with high separative and anbiofouling performance and the resultant membrane exhibits great promise for softening and treatment of wastewater.

## Figures and Tables

**Figure 1 nanomaterials-13-02885-f001:**
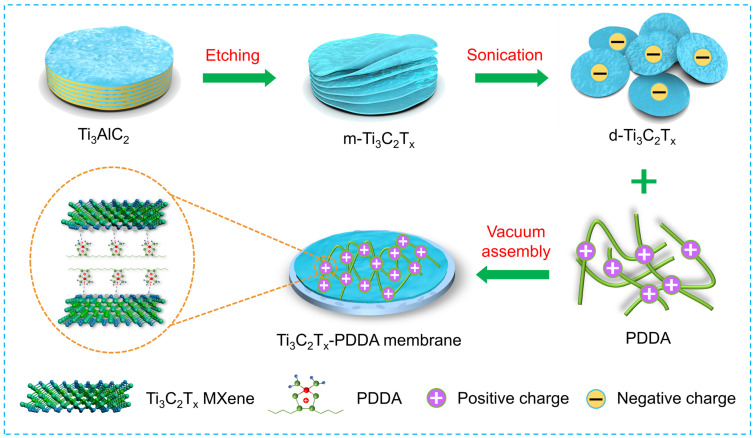
Schematic diagram of the fabrication of Ti_3_C_2_T_x_/PDDA hybrid membranes.

**Figure 2 nanomaterials-13-02885-f002:**
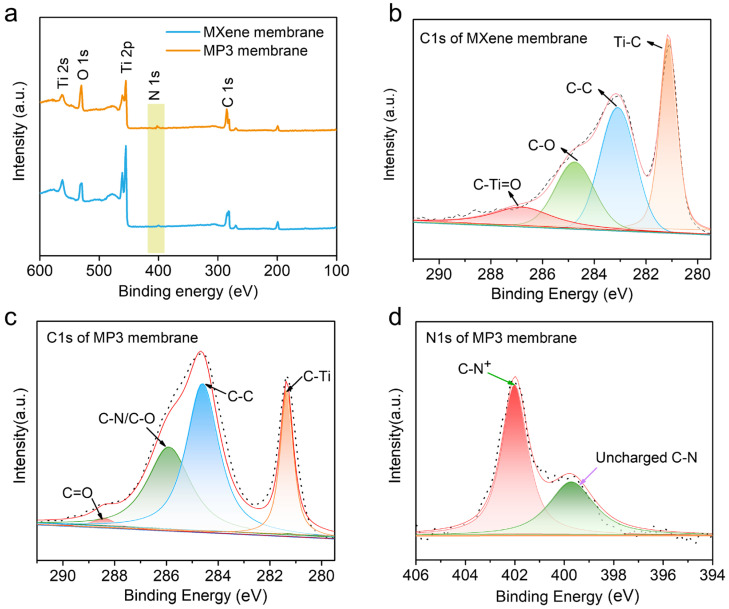
(**a**) XPS survey scan of MXene and MP3; (**b**,**c**) C1s XPS spectra of MXene and MP3; (**d**) N1s XPS spectra of MP3.

**Figure 3 nanomaterials-13-02885-f003:**
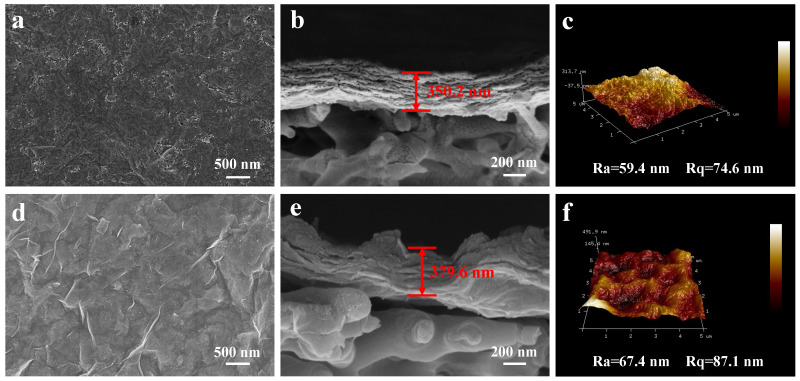
The surface and cross-section SEM images for (**a**,**b**) MXene membrane and (**d**,**e**) MP3 membrane; The 3D AFM images for (**c**) MXene membrane and (**f**) MP3 membrane.

**Figure 4 nanomaterials-13-02885-f004:**
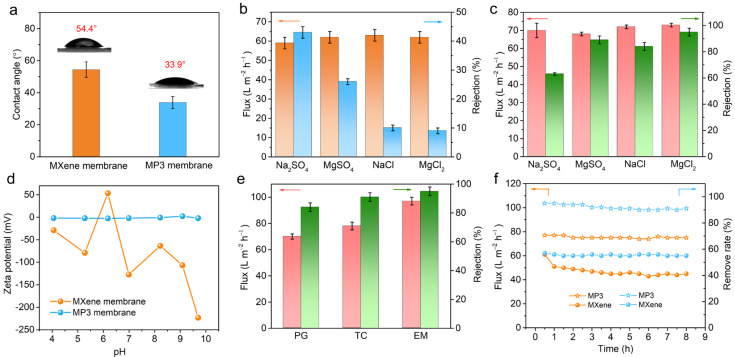
(**a**) Water contact angles of pure Ti_3_C_2_T_x_ membrane and Ti_3_C_2_T_x_/PDDA membrane. Inorganic salt separation behaviors of (**b**) Ti_3_C_2_T_x_ MXene membrane and (**c**) Ti_3_C_2_T_x_/PDDA membrane. (**d**) Zeta potentials of pure Ti_3_C_2_T_x_ membrane and Ti_3_C_2_T_x_/PDDA membrane ranging from the pH of approximately 4.0 to 10.0. (**e**) Antibiotics separation behaviors of Ti_3_C_2_T_x_ membrane. (**f**) Time-dependent separation property of the pure Ti_3_C_2_T_x_ membrane and Ti_3_C_2_T_x_ membranes towards 250 ppm tetracycline as the feed solution at pH = 7.

**Figure 5 nanomaterials-13-02885-f005:**
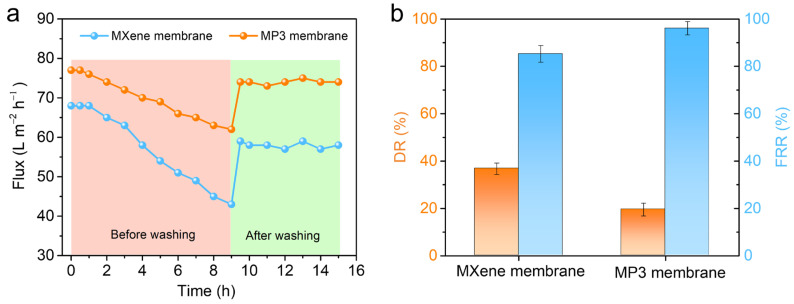
(**a**) Antifouling test of the MXene and MP3 membranes, (**b**) Antifouling index of the MXene and MP3 membranes.

**Figure 6 nanomaterials-13-02885-f006:**
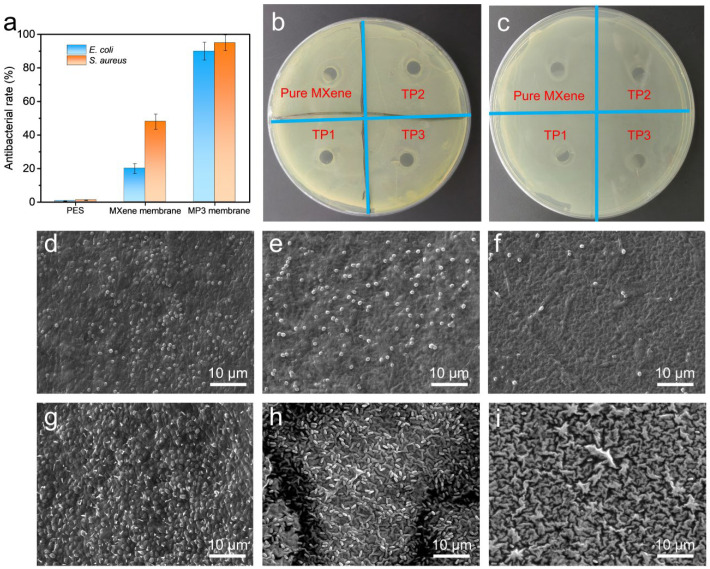
(**a**) Inhibition rate of PES, MXene and MP3 membranes. (**b**,**c**) Antibacterial circle experiment of *S. aureus* and *E. coli* for different membranes. SEM images of the *S. aureus*, respectively, grown on (**d**) PES (control), (**e**) MXene membrane, and (**f**) MP3 membranes surface and the *E. coli* colony grown on (**g**) PES (control), (**h**) MXene membrane, and (**i**) MP3 composite membranes surfaces.

## Data Availability

Data are contained within the article or Appendix A.

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
