# Peer review of "Polycation-Intercalated MXene Membrane with Enhanced Permselective and Anti-Microbial Properties"

_nanomaterials, 2023, doi:10.3390/nano13212885_

Round 1

Reviewer 1 Report

In this article J. Yang et.al, demonstrated the study of permselective and anti-microbial properties based on PDDA/MXene membrane using a facile and patternable electrostatic assembly strategy.  The manuscript was well written and the concept theme is good. The as-prepared PDDA/MXene layered membrane exhibits highest water permeance up to 73.4 L m-2/h with a rejection above 94.6% for MgCl2 with a good swelling resistance and long-term stability and good antimicrobial activity towards E. coil and S. aureus with an inhibition rate of 90 and 95%. The results presented in the study are good and can be likely to attract the readers in in the field of antimicrobial based water purification applications. However, there are some minor  issues that needs to be clarified before the publication, and here are some of my concerns as given below.

1.      The sentence “development of humanity” will not be suitable, please change the word humanity to the suitable word.

2.      Is it “idea choice” or ideal choice? Please check.

3.      There are some other 2D materials like g-C3N4 which is easy to synthesize and have been used in water separation membranes. How does hexene is differ from these materials. Also, please enhance the usage of MXene by insisting any other literature study to support the MXene over the 2D materials.

4.      Please supplement the pristine SEM/XPS wide scan spectra micrograph of the PDDA membrane. How about the O1s peak in XPS. Do the authors observe any peak shifts in the XPS elements after the functionalization of PDDA over MXene layers? If so please write a few lines of its significance and adbot it to your study results for better understanding of your good experiment approach for water purification.

5.      Provide FTIR analysis to support the successful formation of MXene/PDDA nanocomposite.

6.      Please supplement the conductivity of each material.

7.      Some of the references are  not up to the mark, a few of the latest reports are suggested to cite at the 2D materials and MXene related discussions, such as 10.1016/j.foodcont.2022.109022; Science of The Total Environment 811 (2022): 152280. 10.1021/acsmaterialslett.3c00698, Separation and Purification Technology 292 (2022): 121037, 10.1016/j.snb.2023.134471.

Reviewer 2 Report

Introduction needs to be improved, cite recent works with applications (1.      Bacterial adhesion on orthopedic implants. Advances in colloid and interface science. 2020, 283,  1-12)

All graphs need to have experimental errors.

One would expect to have full surface characterization (Charge, hydrophobicity roughness...)

What about advancing /receding angles?

The discussion in terms of surface characteristics is missing!

Reviewer 3 Report

The authors fabricated polydiallyldimethylammonium chloride (PDDA)-intercalated MXene membrane for the nanofiltration application. While the topic is interesting there are some concerns related to the hypothesis of this study and the obtained results that must be addressed.

-          The hypothesis behind this study is not clear. What properties of MXene and PDDA have convinced the authors to use their mixture for this certain application: water softening. PDDA obviously endows positive charge and MXene is highly negatively charged, particularly at high pH values. What is the point of mixing them and coating?

-          TEM cross-sectional images will give more insightful results.

-          It seems that the surface roughness of membranes increased slightly by adding PDDA in MP3. Th reduction in contact angle can be partially attributed to the increased roughness (based on Wenzel, New insights into the impact of nanoscale surface heterogeneity on the wettability of polymeric membranes). However, the main reason for the CA reduction must be the change in the chemistry of coating layer.  Why the presence of PDDA reduces the CA?

-          One of the major challenges of surface-modified membranes is the stability of adhered materials to the surface. LBL strategy makes sure that the adhesion is strong. What strategy the authors followed to improve this attachment of MXene in the coating layer and what are the tests to prove it? See this paper: Zhu et al. Robust superhydrophilic and underwater superoleophobic membrane optimized by Cu doping modified metal-organic frameworks for oil-water separation and water purification

-          Following the previous comment, no results were reported to show the leaching rate of these nanomaterials. These papers might help: In-Situ Ag MOFs Growth on Pre-Grafted Zwitterions Imparts Outstanding Antifouling Properties to Forward Osmosis Membranes and Thermally resistant and electrically conductive PES/ITO nanocomposite membrane.

-          Following the previous comment and the one related to the stability of coated materials, the authors should comment on the production of secondary waste due to the release of MOF

-          The authors need to compare their performance results with the literature. I believe commercial NF membranes made by interfacial polymerization reaction between PIP and TMC provide higher values of flux and rejection. Hence, I cannot comment if the results are promising. Please check these refs: Loose nanofiltration membranes functionalized with in situ-synthesized metal organic framework for water treatment, Micropatterned Thin-Film Composite Poly (piperazine-amide) Nanofiltration Membranes for Wastewater Treatment

-          Following the previous comment, based on the complicated synthesis route of the nanomaterials and coating layer compared to poly (piperazine-amide) nanofiltration membranes, one might question the practicality of this research in real application.

-          Antibacterial behaviors must be proved with some standard tests like inhibition zone, count of colony forming units, confocal microscopy, and SEM of dead and live bacteria; some of them are done in this paper. Also, more details are needed in the Experimental section. These papers might help: Toward Sustainable Tackling of Biofouling Implications and Improved Performance of TFC FO Membranes Modified by Ag-MOFs Nanorods and In-Situ Ag-MOFs Growth on Pre-Grafted Zwitterions Imparts Outstanding Antifouling Properties to Forward Osmosis Membranes and Effective strategy for UV-mediated grafting of biocidal Ag-MOFs on polymeric membranes aimed at enhanced water ultrafiltration

-          The authors need to justify the FRR results based on three parameters: surface roughness, wettability, and charge. Membranes with higher hydrophilicity, more negative charge, and less roughness are typically less prone to fouling (see this work: Treatment of an in situ oil sands produced water by polymeric membranes)

Round 2

Reviewer 1 Report

Authors have made substantial efforts to improve their manuscript standard, and have answered all the quires well. Now this manuscript is much more attractive and better, which may gain positive insights for the readers, and hence this manuscript can be accepted in Nanomaterials journal.

Reviewer 2 Report

can be accepted

Reviewer 3 Report

The comments are applied properly.